# Performance Control in Early Exiting to Deploy Large Models at the Same Cost of Smaller Ones

## Abstract

Early Exiting (EE) is a promising technique for speeding up inference at the cost of limited performance loss. It adaptively allocates compute budget to data points based on their difficulty by exiting at earlier layers when predictions are confident. In this study, we first present a novel perspective on the EE approach, demonstrating that larger models, when deployed with EE, can achieve higher performance than smaller models while maintaining similar computational costs. As existing EE approaches rely on confidence estimation at each exit point, we further study the impact of overconfidence on the controllability of the compute/performance trade-off. We introduce *Performance Control Early Exiting* (PCEE), a method that enables accuracy thresholding by basing decisions not on a datapoint's confidence but on the average accuracy of samples with similar confidence levels from a held-out validation set. In our experiments with MSDNets and Vision Transformer architectures on CIFAR-10, CIFAR-100, and ImageNet, we show that *PCEE* offers a simple yet computationally efficient approach that provides better control over performance than standard confidence-based approaches, and allows us to scale up model sizes to yield performance gain while reducing the computational cost.

## 1 Introduction

Scale, both in terms of model size and amount of data, is the main driver of recent AI developments, as foreseen by Kaplan et al. (2020) and further evidenced by Hoffmann et al. (2022). Remarkably, even model architectures are designed to enable scaling, such as the standard Transformer (Vaswani et al., 2017) which was built to maximize parallelization, facilitating the training of very large models. Similarly, recent recurrent architectures such as RWKV (Peng et al., 2023), Mamba (Gu & Dao, 2023), and xLSTM (Beck et al., 2024) enable scaling for the otherwise inefficient legacy recurrent architectures (Greff et al., 2016) that require sequential processing (Dehghani et al., 2018). The improved prediction performance unlocked with scale unfortunately comes at high memory footprint and latency at inference. Several approaches have been proposed to tackle these limitations, namely quantization (Dettmers et al., 2022; Ma et al., 2024; Dettmers et al., 2024), knowledge distillation (Hinton et al., 2015; Gu et al., 2023; Hsieh et al., 2023) and speculative decoding (Leviathan et al., 2023; Chen et al., 2023) (although specifically for autoregressive models). These methods trade performance for reduced computational cost across all samples, irrespective of their difficulty, with the exception of speculative decoding which uses adaptive computation. However, the speed-up gains from this method are bounded by the quality of the draft model used for speculating predictions. More discussion on related work can be found in Section 6.

In this work, we focus on *Early-Exiting* (EE), an inference optimization technique that allocates budget adaptively to the test samples, based on their perceived difficulty. Early-exit strategies (Grubb & Bagnell, 2012; Huang et al., 2017; Elbayad et al., 2019a; Schuster et al., 2021; Chen et al., 2023) involve establishing exit points at intermediate layers of a network based on the confidence levels of the predictions at each layer. The most common approach within these strategies is to make predictions at each intermediate layer and evaluate their confidence, allowing the model to exit early if the confidence exceeds a predetermined threshold. Figure 2 shows the potential compute savings

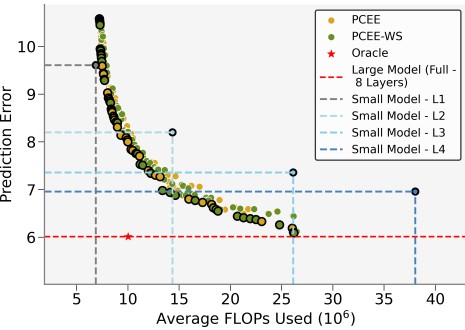 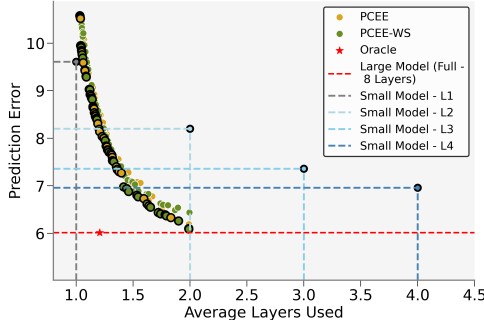

Figure 1: **Larger models coupled with early exiting can achieve lower prediction errors for the same computational budget compared to smaller models.** This plot shows prediction error (%) versus average flops used (left) and average layers used (right) for different MSDNET sizes on CIFAR-10: small (4 layers) and large (8 layers). Various exiting strategies are compared: ours (PCEE, PCEE-WS) and Oracle (exiting as soon as a layer's prediction matches that of the final layer). Each green and yellow dot corresponds to a model seed and a threshold $\delta$. Oracle is computed by averaging over 3 seeds. The large model with any early-exiting strategy gets to lower prediction errors than the full small model with even less compute.

achievable with an Oracle EE strategy that exits at the first layer whose prediction matches that of the last layer.

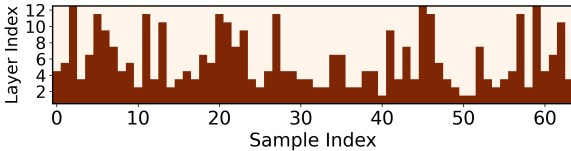

Figure 2: Heatmap of the layers used by an Oracle EE strategy of a VIT on $64$ random samples from IMAGENET-1K. The dark bars indicate the layers used for each sample and the **light-colored area shows the amount of compute that can be saved without losing performance.**

While Early Exiting is commonly used to speed up inference at the cost of performance, in this paper we present a novel perspective by demonstrating that we can achieve the low computational cost of small models and the high performance of large models simultaneously, by training and applying Early Exiting on the large model. In other words, our findings suggest that scaling up models and applying EE is advantageous for both performance and computational efficiency, as depicted in Figure 1 (we observe similar results across several architectures and datasets, as reported in TODO). To achieve such results, performance control is of the essence, *i.e.*, reliably estimating the accuracy of an intermediate prediction so that the model is not prematurely exited. Current EE methods that rely on confidence estimates at each exit point in a multi-layer model are however bound to fail as neural networks are typically miscalibrated (Guo et al., 2017; Wang et al., 2021). To address this, we introduce PCEE, *Performance Control Early Exiting*, a method that ensures a lower bound on accuracy by thresholding based not on a datapoint's confidence but on the average accuracy of its nearest samples from a held-out validation set with similar confidences. This approach offers a simple yet computationally efficient alternative that provides control over performance, facilitating accurate adaptation of EE methods for practical use.

Moving from confidence thresholding to accuracy thresholding has a number of advantages. Unlike confidence, accuracy is an indicator of the actual model performance, hence one can easily decide on to determine a threshold. Confidence estimates can also present inconsistent behavior throughout layers, hence requiring the selection of a different threshold per layer, which is in itself a difficult problem to solve. As discussed in more detail in Section 4 and empirically demonstrated in Section 5, accuracy thresholds offer a simple approach to determine the earliest exit point that guarantees at least the desired accuracy.

**Contributions** Our contributions are summarized as follows:

- We introduce a *post-hoc* early-exit approach called *Performance Control Ealy Exiting (PCEE)* to provide control over accuracy for any model returning a confidence score and a classification decision at each exit point, regardless of how well-calibrated it is.

- Our early exit method requires selecting one single threshold for all layers, unlike existing early exit methods that require learning a threshold per layer. This threshold is a simple accuracy lower bound—based on the target accuracy level chosen by the user—rather than an abstract confidence level unrelated to prediction performance.

- For the first time to our knowledge, we show that scale can also yield inference efficiency. That is, larger models require a reduced amount of computation to attain a certain accuracy level by exiting at very early layers, more so than a smaller model.

## 2 BACKGROUND AND SETTING

We focus on the $K$-way classification setting where data instances correspond to pairs $x, y \sim \mathcal{X} \times \mathcal{Y}$, with $\mathcal{X} \subset \mathbb{R}^d$ and $\mathcal{Y} = \{1, 2, 3, ..., K\}$, $K \in \mathbb{N}$. Classifiers then parameterize data-conditional categorical distributions over $\mathcal{Y}$. That is, a given model $f \in \mathcal{F} : \mathcal{X} \mapsto \Delta^{K-1}$ will project data onto the probability simplex $\Delta^{K-1}$.

**Early Exit Neural Networks** Early Exit Neural Networks enable dynamic resource allocation during model inference, reducing computational demands by not utilizing the entire model stack for every query. These approaches strategically determine the *exit point* for processing based on the perceived difficulty of the data, allowing for a reduction in resource use for simpler examples while allocating more compute power to more complex cases. This is commonly achieved through a confidence threshold $\delta \in [0, 1]$, where the decision to exit early is made if the confidence measure $c_i(x)$ at a given layer $i$—often derived from simple statistics (*e.g.*, $\max(\cdot)$) of the softmax outputs—exceeds $\delta$. While seemingly effective, confidence thresholding is brittle, as it is sensitive to miscalibration, and requires extensive search on a left-out validation dataset to find optimal per-layer thresholds. For example, without properly tuned thresholds, overconfident exit layers result in premature predictions, hence degraded accuracy. We provide a simple fix to this issue in Section 4.

**Calibration and Expected Calibration Error (ECE)** Calibration in multi-class classifiers measures how well the predicted confidence levels (*e.g.*, $\max \text{softmax}(\cdot)$) match the true probabilities of correct predictions (Guo et al., 2017; Nixon et al., 2019). A well-calibrated model means that if a model assigns a 70% confidence to a set of predictions, then about 70% of these predictions should be correct. The Expected Calibration Error (ECE) Naeini et al. (2015) quantifies model calibration by calculating the weighted average discrepancy between average confidence and accuracy across various confidence levels. The formula divides confidence ranges into bins and computes the absolute difference in accuracy and confidence per bin, with an ECE of zero indicating perfect calibration. The formal definition of ECE is available in Section B in the appendix. Reliability diagrams visually assess calibration by comparing confidence levels against actual accuracy in a plot, where deviations from the diagonal ($y = x$) show miscalibration. Overconfidence occurs when confidence exceeds accuracy, while underconfidence happens when it falls short. We will use these reliability diagrams to map confidence to accuracy as discussed in Section 4.2.

## 3 BENEFITS OF INCREASING MODEL SIZE COUPLED WITH EARLY EXITING

Our first contribution is to show that Early Exiting does not necessarily compromise performance for faster inference, but can be used to run larger models at the cost of smaller ones. Figure 1 provides compelling evidence in support of the observation that larger models can lead to greater inference efficiency. Green and yellow dots indicate test error and average FLOPs on the left plot and average layers on the right plot used by EE using the specified method. The prediction error of each layer of the small model (without early exiting) is also shown. The results demonstrate a clear trend: larger models achieve lower prediction errors with fewer FLOPs compared to smaller models if we use early exiting. For instance, the large model with PCEE (our method) achieves a prediction error of around $6\%$ using 2 layers (approximately $26 \times 10^6$ FLOPs) on average. In contrast, the smaller

model utilizing the same amout of FLOPs has a higher level of prediction error (about 7.4%). This difference highlights that larger models can make accurate predictions earlier in the network for most samples, thus saving computational resources on average.

Table 1: Top row shows the **accuracy** (%) of MSDNET small using the full capacity of the model on three different datasets: CIFAR-10, CIFAR-100 and IMAGENET-1K. The bottom row shows the accuracy we can get from MSDNET Large using our EE strategies (PCEE, PCEE-WS) with the **same or less computational cost as the full small model**.

| MSDNET | CIFAR-10 | CIFAR-100 | IMAGENET-1K |
|---|---|---|---|
| Full Small model | 93.04 | 71.24 | 70.7 |
| Large Model with EE | **93.88** | **73.06** | **72.13** |

This observation underscores a significant insight: scaling up model size can enhance computational efficiency by enabling early exits in the inference process. Larger models can leverage their deeper architecture to make correct predictions at earlier stages for easy samples, while benefiting from later layers for hard ones, reducing the need for extensive computation across all layers for all samples. This efficiency is crucial for practical applications, where computational resources and time are often limited. Therefore, our findings challenge the conventional view that larger models are inherently more computationally expensive. Instead, we show that larger models can be more efficient in terms of accuracy for a fixed compute budget, providing a compelling case for scaling up models to improve inference computational efficiency while maintaining or even enhancing prediction accuracy. Table 1 summarizes this observation for CIFAR-10, CIFAR-100, and IMAGENET-1K by showing that the large model with EE can achieve higher performance at the same cost (in FLOPs) of the small model. The inference efficiency plots for these datasets are available in Figures 8 and 9 in the Appendix for prediction error versus both average layers used and average FLOPs used.

Finally, note that these compute gains also translate to reduced latency when using dynamic batching, so that inference is batchified (as for any model without EE) and resources are used at full capacity. Indeed, techniques such as on-the-fly batching (Neubig et al., 2017)[1] enable dynamic batching during inference, allowing the system to start processing new requests as soon as other requests in the batch are completed.

## 4 PERFORMANCE CONTROL EARLY EXITING

In this section, we first examine the miscalibration of Early Exit Neural Networks, demonstrating through experiments that they tend to be overconfident, with miscalibration escalating as layer depth increases. Then we introduce *PCEE (Performance Control Early Exiting)*, a method that ensures a lower bound on accuracy by thresholding not on the confidence estimate of a given test example, but on the average accuracy of samples with similar confidence from a held-out dataset. Our early exit method requires selecting a single threshold rather than one per layer. This threshold is a simple accuracy lower bound, based on the target accuracy chosen by the user, rather than a confidence level that might not relate directly to prediction performance. We emphasize this advantage by highlighting that selecting a threshold per layer involves an exhaustive search over a large space as in existing methods (Elbayad et al., 2019b). For instance, with a 8-layer model, searching for the best threshold for each layer to maximize validation accuracy, even within a narrow range of $(0.8, 0.9]$ and a step size of $0.01$, results in $10^8$ combinations. This extensive search, performed before inference, demands significant computational resources. Additionally, if we need to adjust for lower accuracy due to budget constraints, the entire process must be repeated. In contrast, our method allows easy adjustment of the threshold based on the desired accuracy level, offering significant computational savings and flexibility.

### 4.1 CHECKING FOR MISCALIBRATION IN EARLY EXIT NEURAL NETWORKS

Performing EE at inference to allocate adaptive computation to unseen data requires reliable confidence estimation at each exit point in a multi-layer model. However, this is non-trivial to achieve

---

[1]NVIDIA TensorRT provides libraries to accelerate and optimizer inference performance of large models: https://developer.nvidia.com/tensorrt

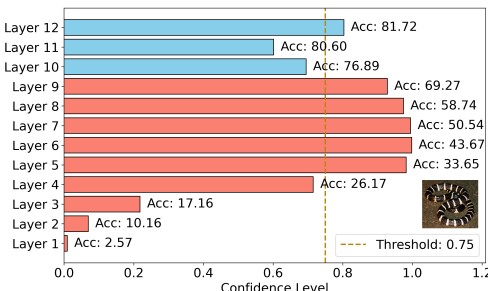

Figure 3: Confidence levels across different layers of a VIT with layerwise classifiers trained on IMAGENET-1K tested on the visually simple snake image shown on the plot. Red bars indicate layers that made incorrect predictions, while blue layers indicate layers that made correct predictions. Overconfident early layers trigger a (premature) exit on layer 5, the first layer surpassing the threshold of 0.75. The test accuracy for each layer is also shown.

as it's well-known that neural networks are typically overconfident (Wang, 2023; Guo et al., 2017). That is, simply relying on commonly used confidence indicators would trigger very early exits at a high rate, damaging overall model performance. Moreover, commonly used confidence estimates are typically somewhat abstract quantities, decoupled from metrics of interest such as prediction accuracy, and it's not easy to decide on confidence thresholds that guarantee a certain performance metric of interest. Jiang et al. (2018) highlights that the model's reported confidence, e.g. probabilities from the softmax layer, may not be trusted especially in critical applications such as medical diagnosis.

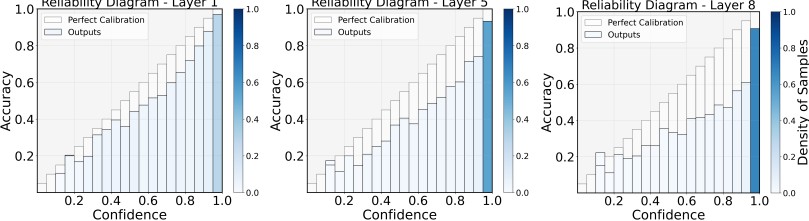

Figure 4: Reliability Diagrams for Layers 1, 5, 8 of MSDNET-LARGE with 8 layers on CIFAR-100

Indeed, if one considers the VIT (Dosovitskiy et al., 2020) with multiple classifiers (*i.e.*, one classifier or *exit point* per layer) trained on IMAGENET-1K (Deng et al., 2009) illustrated in Figure 3, the overconfidence issue becomes noticeable.[2] In the simple example image displayed on the plot, which does not contain distracting objects or a complex background, a confidence threshold of 0.75 would result in a premature exit since early layers are too confident even when wrong, resulting in misclassification. This suggests that accurate exit strategies must be designed. Figure 10 in Appendix D shows a similar phenomenon for MSDNet-Large on an example of CIFAR-100.

Table 2: MSDNET-LARGE on CIFAR-100: Accuracy and ECE of exit points at each of the 8 layers

| Layer | 1 | 2 | 3 | 4 | 5 | 6 | 7 | 8 |
|---|---|---|---|---|---|---|---|---|
| Accuracy (%) | 65.08 | 66.59 | 69.24 | 71.67 | 73.01 | 74.17 | 74.68 | 74.92 |
| ECE | 0.062 | 0.083 | 0.089 | 0.091 | 0.107 | 0.102 | 0.119 | **0.139** |

We further evaluated how commonly used models behave layerwise in terms of overconfidence. To do so, we trained models of varying sizes on CIFAR-10 and CIFAR-100 while adding exit points at every layer. A subset of these results is shown in the reliability diagrams in Figure 4 for certain layers of a MSDNET-LARGE (Huang et al., 2017) with the confidence given the maximum

---

[2]The VIT backbone (without layerwise classifiers) used here is vit_base_patch32_clip_224.laion2b_ft_in1k from TIMM (Wightman, 2019).

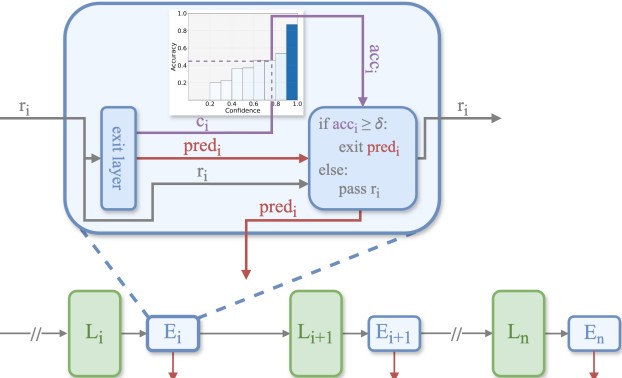

Figure 5: **PCEE**: The structural overview of PCEE. In a multi-layer model with exit points at each layer, the input representation $r_i$ is processed through an exit layer block $E_i$. The exit layer calculates a confidence score $c_i$ and uses a reliability diagram (confidence-to-accuracy mapping) to determine whether to exit or continue processing. If the estimated accuracy from the reliability diagram exceeds an accuracy threshold $\delta$, the model exits and outputs prediction $pred_i$; otherwise, it proceeds to the next layer, passing the representation forward.

of the softmax outputs at each exit point. Additional results with the VIT architecture are shown in Appendix D. Perfectly calibrated models would be such that the bars would hit the $y = x$ line. However, the evaluated model deviates from that, especially so for deeper layers. Table 2 presents ECE for each layer, which increases with depth as already noted from the reliability diagrams.

Also, as discussed in Appendix D, MSDNET-LARGE demonstrates a higher level of overconfidence than MSDNET-SMALL which supports results by Wang (2023) showing that increasing the depth of neural networks increases calibration errors.

## 4.2 PERFORMANCE CONTROL EARLY EXITING (PCEE)

We now introduce PCEE, a method to gain control over performance in Early Exit Neural Networks. The method is illustrated in Figure 5. For a multi-layer model with $n$ layers $\{L_i\}_{i=1}^n$, we incorporate exit points at the end of each layer. At any layer $i$, the input representation of sample $x$ is processed through an exit layer block, denoted as $E_i$, which determines whether the model should terminate at this stage or continue. The exit layer $E_i$ transforms the representation $r_i = L_i(x)$ into a vector of size corresponding to the number of classes.

At this step, a confidence score, $c_i$, for sample $x$, is computed. This score can be derived either as the maximum value or the entropy of the probability distribution obtained after applying softmax. The decision to exit at this layer is then based on the confidence score. As discussed, existing methods rely only on the confidence score itself, which reduces control over accuracy because of the miscalibration issue. To make this decision, we instead employ the reliability diagram for layer $i$, which is constructed from the validation dataset. This diagram provides an estimate of the average accuracy for samples with a confidence level similar to $c_i$ at layer $i$. Suppose $c_i$ falls into bin $m$ of the reliability diagram for layer $i$. If the accuracy corresponding to bin $m$ exceeds a predefined threshold $\delta$, the model exits at layer $i$, outputting the prediction derived from the exit layer. Otherwise, the model proceeds to the next layer. The representation passed to layer $i + 1$ is $r_i$, the one produced at the end of layer $i$ before it goes through $E_i$. Further details of PCEE are outlined in Algorithm 1.

**PCEE-WS** PCEE-WS is a variant of PCEE *with a smoothing* technique applied to the reliability diagrams of the validation dataset. We observed that some bins in the reliability diagrams could contain very few examples, leading to inaccurate representations of the bin's accuracy. To address this, we smooth the accuracy of each example from a binary value (0 or 1) to the average accuracy of its $H$ nearest neighbors based on confidence scores, where $H$ is a hyperparameter. This smoothing is performed before the binning process. The average of these smoothed accuracies is then used to form the bins for the reliability diagrams. Our experimental results demonstrate that this approach

can yield improvements in the performance of the model during inference. We set $H = 150$ in our experiments and used $50$ bins for the reliability diagrams.

**Implementation and Training details** In practice, we implement the exit layers as fully-connected layers that output logits for a softmax layer. We use the softmax maximum as the prediction and its mass as a confidence estimate for that exit layer. Let $\mathbf{z}_i \in \Delta^{K-1}$ be the softmax outputs from exit layer $i$, $E_i$, for a single data instance $x$ with ground truth one-hot encoded label $\mathbf{y}$. The cross-entropy loss for $E_i$ is given by: $\mathcal{L}_i = -\mathbf{y}^\top \log(\mathbf{z}_i)$, and the total loss $\mathcal{L}$ is the average of the cross-entropy losses across all layers: $\mathcal{L} = \frac{1}{n} \sum_{i=1}^{n} -\mathbf{y}^\top \log(\mathbf{z}_i)$. We jointly train the original model architecture and the exit layers by minimizing $\mathcal{L}$ using Stochastic Gradient Descent for CIFAR-10 and CIFAR-100 and AdamW (Loshchilov & Hutter, 2017) for IMAGENET.

Table 3 shows that the addition of intermediate exit layers has a minimal impact on the performance of the original model. This experiment includes two training setups: one with intermediate exit layers on all layers and minimizing $\mathcal{L}$ as described above, and one without intermediate exit layers, i.e., only one classification head at the end. The results indicate that the performance drop from adding intermediate exit layers is negligible considering the error bars, and there is even the possibility of slight accuracy improvement. Overall, the computational savings achieved through early exiting significantly outweigh the minor variation in accuracy.

Table 3: Accuracy comparison of the last layer of MSDNet models with and without intermediate exit layers, showing minimal impact on performance while training with exit layers.

| MSDNET | CIFAR-10 | CIFAR-100 |
|---|---|---|
| Small without intermediate exit layers | $92.05 \pm 0.11$ | $71.47 \pm 0.40$ |
| Small with intermediate exit layers | $92.3 \pm 0.29$ | $71.23 \pm 0.81$ |
| Large without intermediate exit layers | $94.23 \pm 0.24$ | $74.71 \pm 0.53$ |
| Large with intermediate exit layers | $93.86 \pm 0.13$ | $74.85 \pm 0.10$ |

## 5 EXPERIMENTS

We evaluate PCEE and PCEE-WS on widely used image classification benchmarks, and report performance both in terms of accuracy, and computational efficiency. In all experiments, we use $10\%$ of the training data for the CIFAR datasets and $4\%$ for IMAGENET respectively as held-out validation set to learn the confidence-to-accuracy mappings in reliability diagrams for our method, and the hyper-parameters for the baselines. These portions are standard for validation sets on these datasets. For fair comparison, we run all EE methods with thresholds set to the same value for all intermediate layers.

**Baselines** We compare our methods with four baseline approaches: *Oracle*, *Confidence Thresholding* (referred to as "Confidence" in the tables and figures), the *Laplace approximation* introduced by Meronen et al. (2024), and *Confidence Thresholding with Temperature Scaling* (referred to as "TS+Confidence"). *Oracle* refers to a setting with privileged information whereby exits happen as soon as an intermediate layer's prediction matches that of the final layer, showing the potential compute gain of an optimal exiting strategy. The results of Oracle do not depend on the threshold $\delta$. *Confidence Thresholding* checks the confidence of the prediction; if it is above the threshold, it exits. The *Laplace approximation* is a post-hoc calibration method that does not require retraining, like our approach. It approximates a Bayesian posterior for each exit layer with a multi-variate Gaussian, centered on the deterministic exit layer and with covariance equal to the inverse of the exit layer Hessian. Predictions are then obtained via a Monte Carlo estimate that we perform with sample size equal to 1, and with temperature and prior variance set to their default values, following the released codebase. Finally we compare our method to temperature scaling (Guo et al., 2017), a post-hoc calibration technique that divides logits by a scalar parameter, $T$, before applying softmax. In our implementation, we learn one temperature parameter per layer, starting with $T = 1$, and de-

termine its optimal value using the validation set[3]. The TS+Confidence method applies the learned temperature values to the test data, followed by confidence thresholding for early exiting.

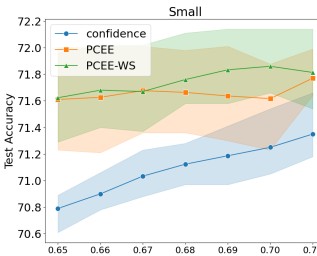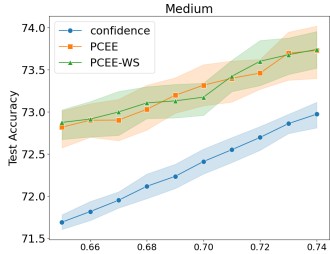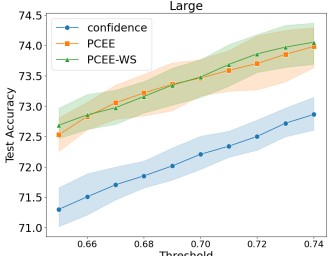

Figure 6: Performance of three MSDNET models (Small, Medium, and Large) evaluated with different thresholds. Each model exits with one of the following methods: *confidence* (blue), PCEE (orange), and PCEE-WS (green). The threshold values correspond to confidence levels that translate to target percentage accuracy. Both PCEE and PCEE-WS methods consistently show higher accuracy than the confidence thresholding, maintaining accuracy above the set threshold. The maximum threshold reflects the peak accuracy achievable by the full model.

**Performance Control** Figure 6 reports results for models of increasing size. We first notice that PCEE (orange) and PCEE-WS (green) show higher *controllability* relative to Confidence Thresholding: resulting accuracy is consistently higher than the threshold for PCEE and PCEE-WS, which is by design and enables simpler inference pipelines where one can compromise accuracy for compute (or vice-versa) more easily than with Confidence Thresholding.

Tables 4 provide a detailed comparison across methods along with computation cost on CIFAR-100. For various threshold values ($\delta$), PCEE and PCEE-WS exhibit higher accuracy compared to baselines. Notably, for the MSDNET-SMALL model, PCEE and PCEE-WS achieve up to 71.81% accuracy at $\delta = 0.71$, outperforming the Confidence's 71.35%. Similarly, PCEE and PCEE-WS reach up to 73.97% accuracy at $\delta = 0.73$ for MSDNET-LARGE, surpassing the Confidence's 72.72% that does not meet the desired threshold. We also highlight that, despite the increase in average number of used layers, PCEE and PCEE-WS achieve higher performance, potentially justifying the computational trade-offs in situations where accuracy is of priority. For example, at $\delta = 0.73$, the MSDNET-LARGE model with PCEE-WS uses 3.02 layers on average, compared to the Confidence's 2.43, reflecting a balance between computational resources and accuracy gains. The Laplace baseline, although using the fewest average layers, falls below the threshold for most of $\delta$ values and therefore does not provide performance control.

**Effect of Calibration** Another finding from Table 4 is that the integration of Temperature Scaling with confidence thresholding enhances performance relative to the confidence baseline. This is an expected result, as TS improves model calibration, and hence accuracy of predictions when early exiting. Still, TS results are slightly worse than those of our proposed PCEE and PCEE-WS methods. It is also important to note that temperature scaling requires additional training and hyperparameter tuning, while our approach offers a simpler alternative that does not necessitate any extra training and remains effective in mitigating overconfidence in models. As highlighted in the related work, model calibration could be a challenging task influenced by various architectural and hyperparameter factors, such as depth, width, and choice of optimizer. Nevertheless, in principle our methods and the TS+Confidence baseline would yield comparable performance with a perfectly calibrated model.

More notably, PCEE and PCEE-WS can also be combined with temperature scaling. As shown in the table, this extra calibration step enhances the performance of our methods, surpassing the other baselines across most thresholds. While our methods are designed to perform well without the need for additional calibration, applying temperature scaling can yield even better results. Therefore, if time and computational resources permit, this step is advisable prior to applying PCEE or PCEE-WS.

---

[3]For the implementation of temperature scaling, we followed https://github.com/gpleiss/temperature_scaling

We observe an interesting result in Table 4, where our methods can achieve higher accuracy than the Oracle while using only a fraction of layers. For example, for the small MSDNet, PCEE-WS can achieve 71.76% accuracy for $\delta = 0.68$ and 71.81% for $\delta = 0.71$, surpassing Oracle's 71.64% accuracy with using only around 50% of the available layers on average. This surprising result can happen when intermediate layers predict the correct label while the last layer does not, known as destructive overthinking (Kaya et al., 2018). This suggests that early exiting (EE) may have a regularizing effect, allowing us to leverage both accuracy and compute efficiency.

Additional results on CIFAR-10 (Figure 13 and Table 7) and IMAGENET (Table 8) are provided in the appendix. Appendix A provides the implementation details of the MSDNET and VIT architectures we use for the experiments throughout the paper.

Table 4: Comparison of EE strategies for MSDNet Small and Large on CIFAR-100. Both PCEE and PCEE-WS consistently show higher accuracy than the other baselines, maintaining accuracy above the set threshold, enabling performance control. Accuracies are averaged over 3 seeds with the standard deviations (std) shown in front of them. The results of Oracle does not depend on $\delta$. Accuracies below the threshold (without considering std) are shown in red. The first and second best accuracies in each row are highlighted in bold. For the small model, PCEE-WS surpasses the oracle accuracy using only about 50% of the available layers on average.

| $\delta$ | MSDNET SMALL | Oracle | Confidence | Laplace | TS+Confidence | PCEE (ours) | PCEE-WS (ours) | TS+PCEE-WS (ours) |
|---|---|---|---|---|---|---|---|---|
| | ACC ↑ | **71.64** | $70.79 \pm 0.16$ | $64.90 \pm 0.49$ | $71.16 \pm 0.19$ | $71.61 \pm 0.39$ | **71.62** $\pm 0.33$ | **71.62** $\pm 0.25$ |
| 0.65 | Avg Layers ↓ | 1.63 | $1.64 \pm 0.01$ | $1.31 \pm 0.05$ | $1.73 \pm 0.01$ | $1.96 \pm 0.02$ | $1.95 \pm 0.02$ | $1.96 \pm 0.05$ |
| | Avg FLOPs ($10^6$) ↓ | 13.02 | $12.92 \pm 0.11$ | $10.01 \pm 0.42$ | $13.89 \pm 0.09$ | $16.30 \pm 0.28$ | $16.16 \pm 0.12$ | $16.24 \pm 0.46$ |
| | ACC ↑ | 71.64 | $71.12 \pm 0.16$ | $64.17 \pm 0.38$ | $71.33 \pm 0.24$ | $71.66 \pm 0.31$ | **71.76** $\pm 0.30$ | **71.78** $\pm 0.39$ |
| 0.68 | Avg Layers ↓ | 1.63 | $1.71 \pm 0.01$ | $1.29 \pm 0.05$ | $1.81 \pm 0.01$ | $2.01 \pm 0.04$ | $2.05 \pm 0.02$ | $2.03 \pm 0.02$ |
| | Avg FLOPs ($10^6$) ↓ | 13.02 | $13.61 \pm 0.11$ | $10.13 \pm 0.43$ | $14.68 \pm 0.1$ | $16.77 \pm 0.44$ | $17.16 \pm 0.30$ | $16.94 \pm 0.2$ |
| | ACC ↑ | 71.64 | $71.35 \pm 0.27$ | $63.26 \pm 0.48$ | $71.55 \pm 0.39$ | $71.77 \pm 0.19$ | **71.81** $\pm 0.30$ | **71.79** $\pm 0.33$ |
| 0.71 | Avg Layers ↓ | 1.63 | $1.78 \pm 0.01$ | $1.27 \pm 0.05$ | $1.88 \pm 0.01$ | $2.10 \pm 0.07$ | $2.10 \pm 0.04$ | $2.10 \pm 0.02$ |
| | Avg FLOPs ($10^6$) ↓ | 13.02 | $14.36 \pm 0.14$ | $9.88 \pm 0.43$ | $15.44 \pm 0.12$ | $17.74 \pm 0.65$ | $17.77 \pm 0.35$ | $17.78 \pm 0.17$ |
| $\delta$ | MSDNET LARGE | | | | | | | |
| | ACC ↑ | **74.9** | $71.70 \pm 0.34$ | $69.41 \pm 0.39$ | $72.62 \pm 0.21$ | $73.05 \pm 0.38$ | $72.97 \pm 0.37$ | **73.11** $\pm 0.14$ |
| 0.67 | Avg Layers ↓ | 2.09 | $2.16 \pm 0.01$ | $1.94 \pm 0.08$ | $2.40 \pm 0.01$ | $2.64 \pm 0.05$ | $2.62 \pm 0.06$ | $2.62 \pm 0.01$ |
| | Avg FLOPs ($10^6$) ↓ | 27.47 | $27.24 \pm 0.37$ | $23.68 \pm 1.07$ | $33.63 \pm 0.34$ | $39.96 \pm 1.03$ | $39.77 \pm 1.37$ | $40.19 \pm 0.89$ |
| | ACC ↑ | **74.9** | $72.21 \pm 0.33$ | $69.18 \pm 0.51$ | $73.09 \pm 0.36$ | $73.46 \pm 0.37$ | $73.48 \pm 0.41$ | **73.54** $\pm 0.27$ |
| 0.7 | Avg Layers ↓ | 2.09 | $2.29 \pm 0.02$ | $1.98 \pm 0.06$ | $2.54 \pm 0.01$ | $2.80 \pm 0.07$ | $2.82 \pm 0.07$ | $2.78 \pm 0.06$ |
| | Avg FLOPs ($10^6$) ↓ | 27.47 | $30.20 \pm 0.43$ | $24.75 \pm 0.89$ | $37.17 \pm 0.43$ | $44.21 \pm 1.76$ | $44.72 \pm 1.65$ | $44.09 \pm 1.73$ |
| | ACC ↑ | **74.9** | $72.72 \pm 0.32$ | $68.80 \pm 0.77$ | $73.62 \pm 0.23$ | $73.85 \pm 0.58$ | **73.97** $\pm 0.46$ | $73.93 \pm 0.36$ |
| 0.73 | Avg Layers ↓ | 2.09 | $2.43 \pm 0.01$ | $2.00 \pm 0.06$ | $2.70 \pm 0.01$ | $2.99 \pm 0.09$ | $3.02 \pm 0.11$ | $2.98 \pm 0.09$ |
| | Avg FLOPs ($10^6$) ↓ | 27.47 | $33.52 \pm 0.37$ | $25.59 \pm 0.85$ | $40.97 \pm 0.35$ | $49.04 \pm 2.08$ | $50.18 \pm 2.55$ | $49.32 \pm 1.99$ |

# 6 RELATED WORK

**Inference Efficiency** Inference efficiency has been tackled in many different ways. For instance, quantization approaches (Dettmers et al., 2022; Ma et al., 2024; Dettmers et al., 2024) reduce the numerical precision of either model parameters or data, although typically at the expense of accuracy. Knowledge distillation approaches (Hinton et al., 2015; Gu et al., 2023; Hsieh et al., 2023) were also introduced with the aim of accelerating inference by training a small model to imitate a large one. While yielding improvements in inference speed, distilled models may miss certain capabilities that only manifest at scale (Wei et al., 2022). A recent line of work, called speculative decoding (Leviathan et al., 2023; Chen et al., 2023), uses instead a small model for drafting a proposal prediction but keeps the large one for scoring and deciding whether to accept or reject it. Although exact, speculative decoding speed-up relies on the quality of the small model used for drafting, as a better drafter results in higher token acceptance rates and longer speculated sequences. Moreover, such techniques are not suited to non-autoregressive models, such as classifiers.

**Early Exit Neural Networks** The first instance of EE was introduced by Teerapittayanon et al. (2016) where exit classifiers are placed after several layers, operating on top of intermediate representations. At training time, the joint likelihood is maximized for all exit points, while at inference the decision of whether or not to exit at each exit point is made by thresholding the entropy of the predicted categorical. This approach suffers from the overconfidence of neural networks, which triggers premature exits. While there exist approaches aimed at improving overconfidence such as nonparametrical TRUST SCORES Jiang et al. (2018) or simply improving the accuracy of the under-

lying classifier (Vaze et al., 2021; Feng et al., 2022), those wouldn't scale to the early-exit setting that requires overconfidence to be tackled for every exit point. Recent work (Meronen et al., 2024) also tackles the overconfidence issue by better estimating uncertainty via a post-hoc Bayesian approach and leveraging model-internal ensembles. This approach is specific to linear exit layers and adds a significant overhead, as it requires estimating for each exit layer its Hessian to approximate a Bayesian posterior and sample from it. Görmez et al. (2021) propose instead an architecture variation that leverages prototypical classifiers (Papernot & McDaniel, 2018) at every layer to avoid training early exit classifiers, at the cost of having to threshold on unbounded distances.

Even for well-calibrated models, challenges persist as they require careful tuning of a threshold per exit point, which is far from trivial and involves mapping abstract confidence measures such as entropy to some performance metric of interest. Ilhan et al. (2024) propose training a separate model parameterizing a policy that decides on exit points. Alternatively, we seek to do so with an efficient non-parametric approach that thresholds on target accuracy levels. We would go as far as to speculate that *the difficulty in selecting thresholds yielding a certain level of performance is the main reason why early exit approaches are not currently widely used in practical applications*. Extensions to the sequence setting were also proposed recently, such as Schuster et al. (2022), but as with any other existing approach, a threshold needs to be picked for every layer, and it's difficult to anticipate the downstream performance for a given choice of the set of thresholds.

**Model Calibration** Confidence estimation plays a central role in EE approaches since calibrated models enable deciding when to early exit by simply comparing confidence levels with user-specified thresholds. However, recent work (Guo et al., 2017) pointed out that neural networks tend to be poorly calibrated despite having high predictive power and achieving high accuracy, and larger models tend to be primarily overconfident (Carrell et al., 2022; Hebbalaguppe et al., 2022; Wang, 2023). Calibrating models is a complex challenge due to the interplay of multiple architectural and hyperparameter factors (Hebbalaguppe et al., 2022). Indeed, recent work showed that the depth, width, weight decay, batch normalization, choice of optimizers and activation functions, and even the datasets themselves significantly influence calibration (Guo et al., 2017; Hein et al., 2018).

## 7 CONCLUSION AND DISCUSSION

We have presented a computationally efficient method for reliably early exiting and showed that we can achieve the accuracy of large models with a fraction of the compute required even by small ones. Our method makes use of a held-out validation set to estimate the mapping from confidence to accuracy in intermediate layers. This provides the user with better control over the model to match a desired accuracy target and simplifies the threshold selection procedure. Compared to confidence thresholding, we have shown that our method consistently improves the final accuracy when applied to models that are overconfident, as typically observed in the literature. We note however that this behavior is not necessarily true for underconfident models, as reported in Appendix E.4. Finally, like when running the original model without EE, our method does not handle out-of-distribution data well and suffers from discrepancies between the validation and test sets (a weakness shared with Temperature Scaling that is shown to be not effective for calibration under distribution shifts (Ovadia et al., 2019; Tada & Naganuma, 2023; Chidambaram & Ge, 2024)). This issue can be mitigated in deployment by continuously updating our reliability diagrams using fresh data as they come to account for distribution shifts over time. Another solution to this problem is to enable rejection (e.g., by adding an "I don't know" class) to make the model more robust to distribution shifts (see for instance Liu et al. (2019)). Studying the compatibility of such an approach with EE is the subject of future work.

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

## A    IMPLEMENTATION DETAILS

**MSDNets**    MSDNets, proposed by (Huang et al., 2017), serve as a benchmark for EENNs (Jazbec et al., 2023; Laskaridis et al., 2021; Han et al., 2022) known for their overconfidence issues (Mero-nen et al., 2024). MSDNet's architectural design addresses significant challenges in incorporating intermediate exits into Convolutional Neural Networks (CNNs). One major challenge is the lack of coarse features in the early layers, which are essential for effective classification. Capturing essential coarse features, such as shape, is critical in the early layers, as classifying based on shape is easier and more robust than using edges or colors. Another challenge is the conflict of gradients arising from multiple sources of loss from the exit layers, which hinders the transmission of rich information to the end of the network. To tackle these challenges, MSDNet incorporates vertical convolutional layers—also known as scales—that transform finer features into coarse features at every layer and introduce dense connectivity between the layers and scales across the network, effectively reduc-ing the impact of conflicting gradients. MSDNets used throughout the paper are in 3 different sizes: Small, Medium, and Large. For CIFAR datasets, they only differ in the number of layers, 4 layers for Small, 6 layers for Medium, and 8 layers for Large. For ImageNet, they all have 5 layers but the base is 4, 6, 7 respectively. For the arguments specific to MSDNets and the learning rate scheduler, we followed the code in this repository: https://github.com/AaltoML/calibrated-dnn. To train the models, we used an SGD optimizer with a training batch size of 64, an initial learning rate of 0.01, a momentum of 0.9, and a weight decay of 1e-4 for CIFAR datasets and AdamW with an initial learning rate of 0.4, a weight decay of 1e-4 and batch size of 1024 for ImageNet.

**ViT**    The ViT (Dosovitskiy et al., 2020) model we used for the experiments on CIFAR datasets is a 12-layer self-attentive encoder with 8 heads, trained with AdamW with a learning rate of 1e-3, a weight decay of 5e-5, a cosine annealing learning rate scheduler and a training batch size of 64. The Vit Small model in Table 9 has the same architecture as the 12-layer larger model but has 6 layers. The evolution of train and test errors through epochs of the last layer of the ViT trained on CIFAR-10 in Figure 11b is plotted in Figure 7. The reliability diagrams were plotted at an epoch where the model demonstrated good generalization performance, characterized by low train error and stabilized test error.

Most of our experiments can be carried out in single-gpu settings with gpus with at most 16 Gb of memory, under less than a day. For ImageNet, training was carried out with data parallelism over 4 32 Gb gpus, which took less than two days.

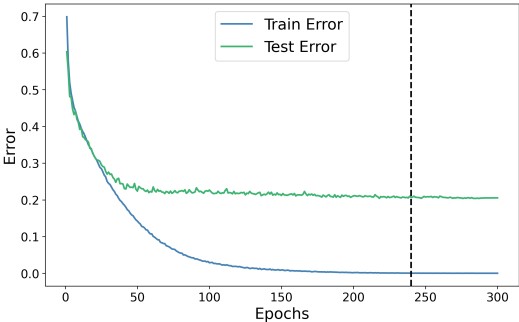

Figure 7: The evolution of train and test errors for ViT on CIFAR-10. The vertical dashed line is where we plotted the reliability diagrams in Fig 11b.

## B    ECE FORMAL DEFINITION

As discussed in the Background Section 2, calibration of a multi-class classifier refers to how well the predicted confidence levels (*e.g.*, $\max \text{softmax}(\cdot)$) match the actual probabilities of correct pre-dictions. In other words, a model is considered well-calibrated if, for any given confidence level, the predicted probability of correctness closely matches the observed frequency of correctness. For example, if a model assigns a $70\%$ confidence to a set of predictions, ideally, approximately $70\%$ of those predictions should be correct. The Expected Calibration Error (ECE) (Naeini et al., 2015) is

often used to quantify the calibration of a model since it measures the weighted average difference between the average confidence and accuracy, across multiple confidence levels. More formally, ECE is defined as follows if we split the range of confidences observed by $f \in \mathcal{F}$ from a sample of data points $X$ into $M$ bins:

$$\text{ECE}(f, X) = \sum_{m=1}^{M} \frac{|B_m|}{n} |acc(B_m, f) - conf(B_m, f)| \tag{1}$$

where $B_m$ is the set of data instances whose predicted confidence scores fall into the $m$-th bin, $acc(B_m, f)$ is the accuracy of the model measured within $B_m$, and $conf(B_m, f)$ is the average confidence of predictions in $B_m$, assuming measures of confidence within the unit interval. An ECE of 0 would indicate a perfectly calibrated $f$ on $X$. Reliability diagrams are visual tools used to evaluate calibration by plotting confidence bins against accuracy. Deviations from the $y = x$ diagonal line in a reliability diagram indicate miscalibration, with overconfidence and underconfidence representing predictions where the model's confidence consistently exceeds or falls short of the actual accuracy, respectively.

## C  PCEE ALGORITHM

Algorithm 1 shows our methodology for performance controllability.

---

**Algorithm 1** Inference with PCEE

---

1: **Require:** Model $A$ with $n$ layers, accuracy threshold $\delta$, reliability diagrams $D$
2: **for** each layer $i = 1$ to $n - 1$ in $A$ **do**
3:     Process input by $L_i$, then pass its output $r_i$ to $E_i$
4:     Compute confidence score $c_i$ from $r_i$
5:     Obtain accuracy $acc_i$ from reliability diagram $D_i$ for $c_i$
6:     **if** $acc_i \geq \delta$ **then**
7:         **exit** and output prediction $pred_i$
8:     **else**
9:         Pass $r_i$ to the next layer $L_{i+1}$
10:     **end if**
11: **end for**
12: **Output** prediction $pred_n$ from the last exit $E_n$

---

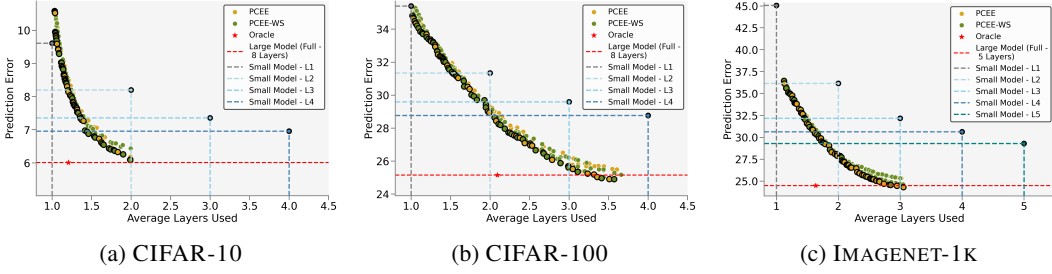

(a) CIFAR-10       (b) CIFAR-100       (c) IMAGENET-1K

Figure 8: Effect of model size on inference efficiency for MSDNet on three datasets: Prediction error (%) vs. average layers used.

## D  ADDITIONAL RESULTS EVALUATING OVERCONFIDENCE

In this section, we provide more experimental details and results to complement those of Section 4.1. Figure 10 shows a similar phenomenon to Figure 3 for MSDNet on a simple example of CIFAR-100. This figure shows that the model surpasses the threshold of 0.75 at the first layer although being wrong until layer 6.

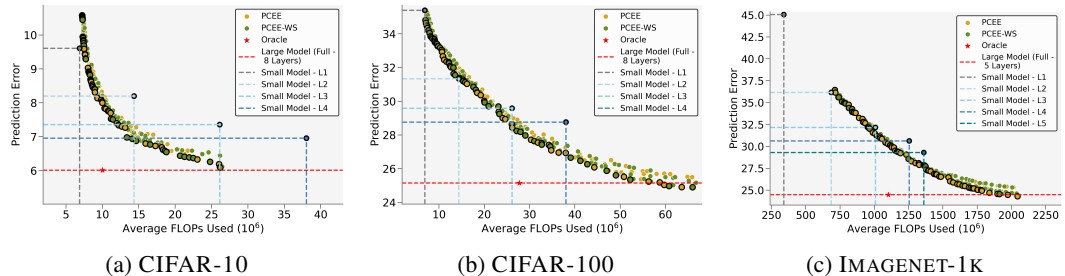

(a) CIFAR-10      (b) CIFAR-100      (c) IMAGENET-1K

Figure 9: Effect of model size on inference efficiency for MSDNet on three datasets: Prediction error (%) vs. average FLOPs used.

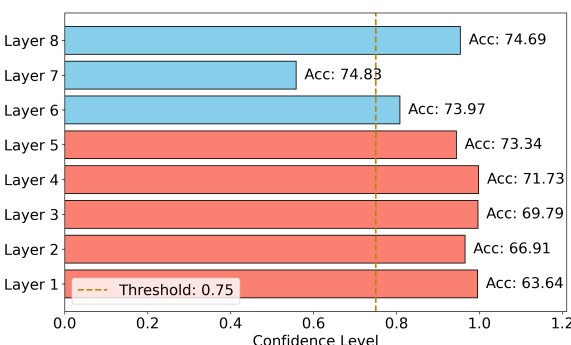

Figure 10: Confidence levels across different layers of a MSDNET with layerwise classifiers trained on CIFAR-100 tested on a random image from this dataset. Red bars indicate layers that made incorrect predictions, while blue layers indicate layers that made correct predictions. Overconfident early layers trigger a (premature) exit on layer 1, the first layer surpassing the threshold of 0.75 although the model makes incorrect predictions until layer 6. The test accuracy for each layer is also shown.

Figure 11 shows the reliability diagrams for MSDNet Large on CIFAR-100 and VIT on CIFAR-10 through different exit layers. The confidence measure here is the maximum softmax output. Results led to the two following observations:

- **Effect of depth:** Calibration degrades and models become overconfident for deeper layers. Table 5 presents ECE for each layer, which increases with depth in both architectures.

- **Effect of model size:** MSDNET-LARGE demonstrates a higher level of overconfidence than MSDNet Small, particularly towards the later layers, which supports the claim in Wang (2023) empirically that increasing the depth of neural networks increases calibration errors (see Figure 12 and Table 6).

For VIT on CIFAR-10 we compare our plots with Carrell et al. (2022). While Carrell et al. (2022) does not provide code for their plots, our results align well with theirs in terms of the reliability diagram for the last layer and the test error.

# E    FURTHER EXPERIMENTAL RESULTS

## E.1    CIFAR-10

For the results of MSDNet on CIFAR-10, refer to Figure 13 and table 7. All methods perform well on this relatively simple dataset, achieving top-1 accuracies above the threshold.

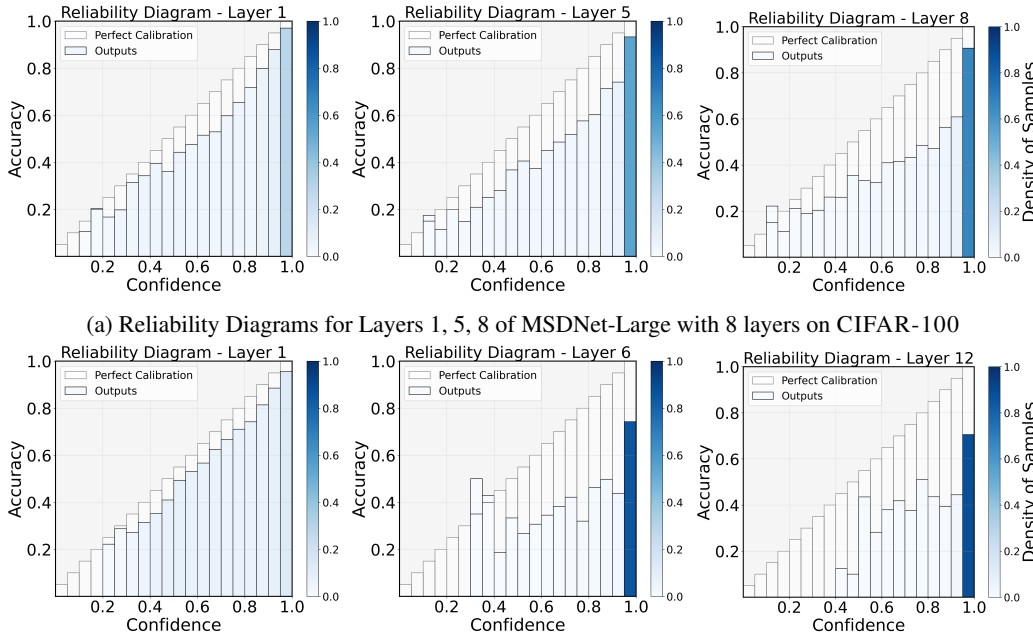

(a) Reliability Diagrams for Layers 1, 5, 8 of MSDNet-Large with 8 layers on CIFAR-100

(b) Reliability Diagrams for layers 1, 6, and 12 of ViT with 12 layers on CIFAR-10

Figure 11: Reliability diagrams for MSDNet-Large and ViT on CIFAR datasets

Table 5: ECE for different layers of the models shown in Figure 11

(a) MSDNet-Large on CIFAR-100: Accuracy and ECE for each of the 8 layers

| Layer | 1 | 2 | 3 | 4 | 5 | 6 | 7 | 8 |
|---|---|---|---|---|---|---|---|---|
| Accuracy (%) | 65.08 | 66.59 | 69.24 | 71.67 | 73.01 | 74.17 | 74.68 | 74.92 |
| ECE | 0.062 | 0.083 | 0.089 | 0.091 | 0.107 | 0.102 | 0.119 | **0.139** |

(b) ViT on CIFAR-10: Accuracy and ECE of each of the 12 layers

| Layer | 1 | 2 | 3 | 4 | 5 | 6 | 7 | 8 | 9 | 10 | 11 | 12 |
|---|---|---|---|---|---|---|---|---|---|---|---|---|
| Accuracy (%) | 62.14 | 72.33 | 75.76 | 77.79 | 78.29 | 78.37 | 78.77 | 78.94 | 79.06 | 79.10 | 79.15 | 79.25 |
| ECE | 0.051 | 0.143 | 0.191 | 0.231 | 0.254 | 0.269 | 0.277 | 0.282 | 0.282 | 0.286 | 0.294 | **0.299** |

## E.2 IMAGENET

In Table 8, we report results for MSDNet Large on ImageNet where our method consistently achieves accuracy higher than the target ones (as indicated by the chosen threshold). Interestingly, in this setting the Confidence baseline also satisfies the control property, and generally results in higher accuracy at the cost of higher compute.

## E.3 BENEFITS OF USING A LARGER MODEL COUPLED WITH EE

Table 9 on CIFAR-10 and CIFAR-100 shows that by using a large VIT model with EE, we can get a better performance than the full small VIT with the same or less computational cost.

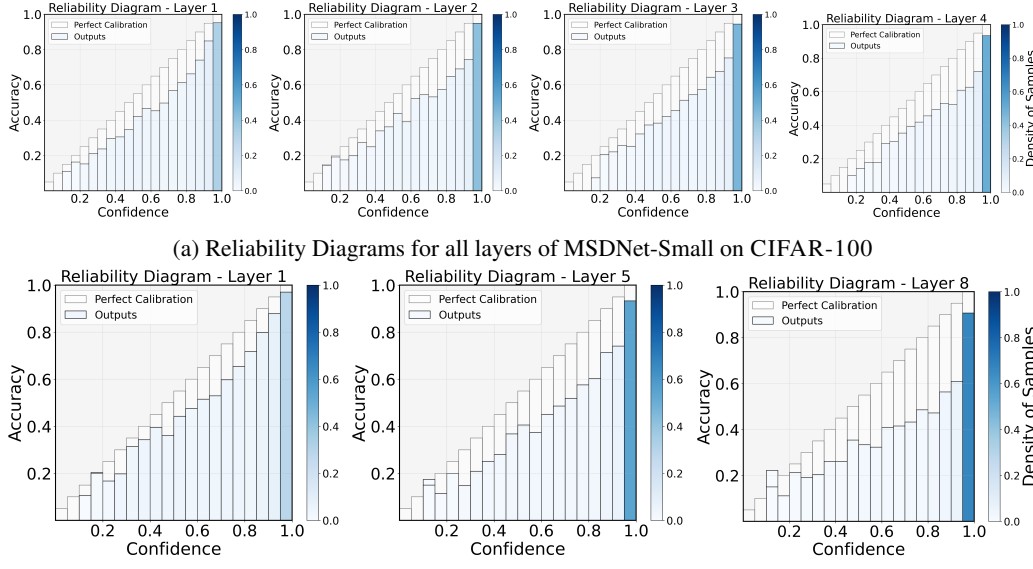

(a) Reliability Diagrams for all layers of MSDNet-Small on CIFAR-100

(b) Reliability Diagrams for layers 1, 5, and 8 of MSDNet-Large on CIFAR-100

Figure 12: Reliability diagrams for MSDNet Small and Large on CIFAR-100

Table 6: MSDNet-Small and Large on CIFAR-100: per-layer Expected Calibration Errors

(a) MSDNet-Small with 4 layers

| Layer | 1 | 2 | 3 | 4 |
|---|---|---|---|---|
| Accuracy (%) | 64.61 | 69.02 | 70.92 | 71.40 |
| ECE | 0.093 | 0.099 | 0.104 | **0.1173** |

(b) MSDNet-Large with 8 layers

| Layer | 1 | 2 | 3 | 4 | 5 | 6 | 7 | 8 |
|---|---|---|---|---|---|---|---|---|
| Accuracy (%) | 65.08 | 66.59 | 69.24 | 71.67 | 73.01 | 74.17 | 74.68 | 74.92 |
| ECE | 0.062 | 0.083 | 0.089 | 0.091 | 0.107 | 0.102 | 0.119 | **0.139** |

### E.4 RESULTS ON AN UNDERCONFIDENT MODEL

The models we tested so far where generally overconfident, which is a typical characteristic of deep learning models. We here report results for a pre-trained ViT model[4], that we observe to be underconfident for most of its layers on ImageNet as shown in Figure 14. In this setting, we observe that the final accuracy of our method is still above the target one, as reported in Figure 15. However, thresholding on confidence achieves higher performance in this particular case, even though consuming more compute. When analyzing the accuracy/compute trade-off in Figure 16, the gap between our method and the baseline are not noticeable, indicating that our method does not degrade performance at the very least. One noticeable difference is the lack of low accuracy/low compute points for the confidence thresholding baseline. Indeed using a confidence estimate that is lower than the actual accuracy (underconfidence) makes the model use more layers to meet the threshold. In contrast, our methods check the accuracy of the bin where the threshold falls and can exit early because its accuracy estimate meets the threshold. Therefore, in Figure 16, we see that our methods can output low accuracies (i.e., higher prediction errors) and show the controllability over low accuracy region that would not be achievable with only confidence thresholding. This confirms the intuition that miscalibration causes different types of problems for early exiting, even in underconfidence scenarios, which are typically not seen.

---

[4]**vit_base_patch32_clip_224.laion2b_ft_in1k** from timm (Wightman, 2019)

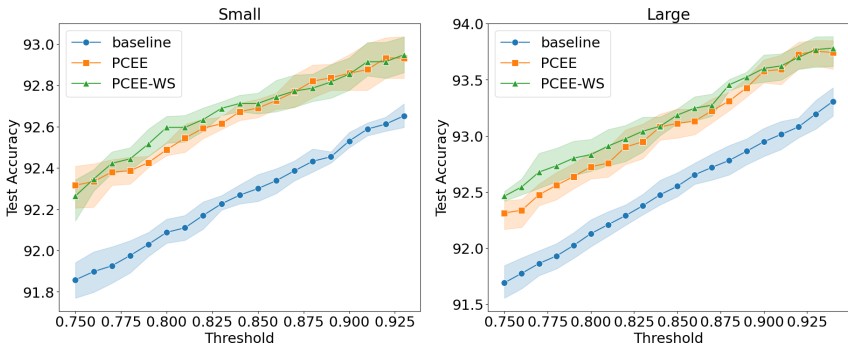

Figure 13: The plot shows the performance of MSDNet Small and Large evaluated with different threshold values on CIFAR-10. Each model's performance is represented by three methods: Confidence baseline (blue), PCEE (orange), and PCEE-WS (green). The threshold values correspond to confidence levels that translate directly to accuracy (accuracy = $100 \times$ threshold). Both PCEE and PCEE-WS methods consistently show higher accuracy than the Confidence baseline, maintaining accuracy above the set threshold. The maximum threshold reflects the peak accuracy achievable by the full model.

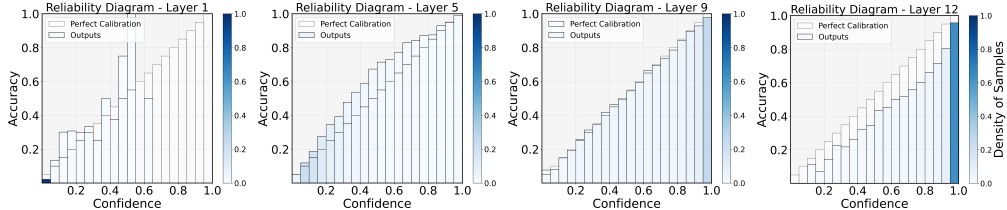

Figure 14: Reliability Diagrams for Layers 1, 5, 9, 12 of ViT with 12 layers on IMAGENET. Early layers are underconfident, and the model smoothly becomes more confident as depth increases, turning overconfident past layer 9.

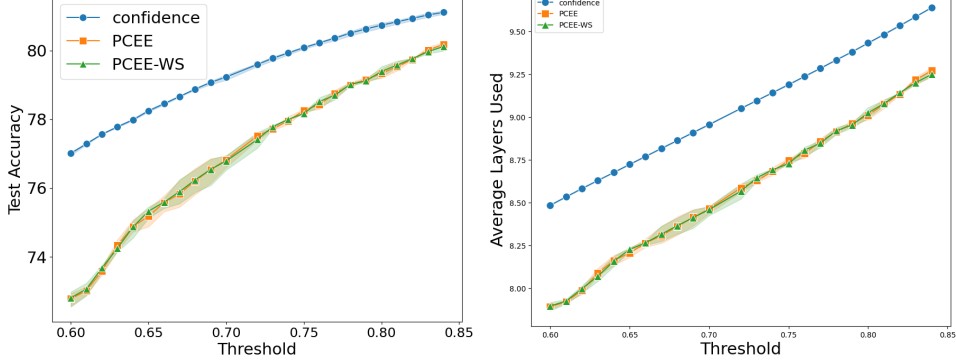

Figure 15: Performance of pre-trained ViT on ImageNet as a function of the selected threshold.

Table 7: This table compares PCEE and PCEE-WS with the Confidence baseline for MSDNet Small and Large on CIFAR-10. Both PCEE and PCEE-WS consistently show higher accuracy than the other methods, maintaining accuracy above the set threshold that fulfills our claim of more controllability on the accuracy.

| | ACC ↑ | Avg Layers ↓ | FLOPs ($10^6$) ↓ |
|---|---|---|---|
| **Small** | | | |
| MSDNet (Oracle) | 92.94 | 1.13 | |
| $\delta = 0.75$ | | | |
| - PCEE (ours) | **92.32** | 1.34 | 10.13 |
| - PCEE-WS (ours) | 92.26 | 1.33 | 9.99 |
| - Confidence | 91.86 | 1.13 | 7.98 |
| - Laplace | 91.09 | 1.13 | 8.07 |
| $\delta = 0.85$ | | | |
| - PCEE (ours) | 92.69 | 1.47 | 11.42 |
| - PCEE-WS (ours) | **92.71** | 1.49 | 11.53 |
| - Confidence | 92.30 | 1.20 | 8.73 |
| - Laplace | 89.89 | 1.15 | 8.37 |
| $\delta = 0.92$ | | | |
| - PCEE (ours) | **92.93** | 1.63 | 12.98 |
| - PCEE-WS (ours) | 92.91 | 1.62 | 12.85 |
| - Confidence | 92.61 | 1.31 | 9.74 |
| - Laplace | 87.40 | 1.15 | 8.48 |
| **Large** | | | |
| MSDNet (Oracle) | 94.04 | 1.21 | |
| $\delta = 0.75$ | | | |
| - PCEE (ours) | 92.31 | 1.34 | 11.88 |
| - PCEE-WS (ours) | **92.46** | 1.35 | 12.16 |
| - Confidence | 91.69 | 1.18 | 9.22 |
| - Laplace | 92.09 | 1.25 | 10.39 |
| $\delta = 0.85$ | | | |
| - PCEE (ours) | 93.11 | 1.57 | 17.00 |
| - PCEE-WS (ours) | **93.19** | 1.60 | 17.65 |
| - Confidence | 92.55 | 1.32 | 11.72 |
| - Laplace | 92.21 | 1.39 | 12.83 |
| $\delta = 0.93$ | | | |
| - PCEE (ours) | 93.75 | 1.95 | 25.30 |
| - PCEE-WS (ours) | **93.77** | 1.96 | 25.63 |
| - Confidence | 93.20 | 1.54 | 16.36 |
| - Laplace | 91.11 | 1.52 | 15.67 |

Table 8: Comparison of EE strategies for MSDNet Large on ImageNet.

| $\delta$ | MSDNET LARGE | *Oracle* | *PCEE* (ours) | *PCEE-WS* (ours) | *Confidence* |
|---|---|---|---|---|---|
| best | **ACC ↑** | **75.51** | 75.25 | 75.25 | 75.41 |
| | **Avg Layers ↓** | 1.63 | 3.03 | 3.03 | 3.23 |
| 0.71 | **ACC ↑** | - | 74.20 | 74.19 | **74.90** |
| | **Avg Layers ↓** | - | 2.48 | 2.48 | 2.71 |
| 0.74 | **ACC ↑** | - | 74.39 | 74.36 | **75.08** |
| | **Avg Layers ↓** | - | 2.56 | 2.55 | 2.82 |

Table 9: Top row shows the **accuracy** (%) of VIT Small using the full capacity of the model on CIFAR-10 and CIFAR-100. The bottom row shows the accuracy we can get from VIT Large using our EE strategies (PCEE, PCEE-WS) with the **same or less computational cost as the full small model**.

| Model | Size | CIFAR-10 | CIFAR-100 |
|---|---|---|---|
| VIT | Full Small model | 88.15 | 62.94 |
|  | Large Model with EE | **90.16** | **63.38** |

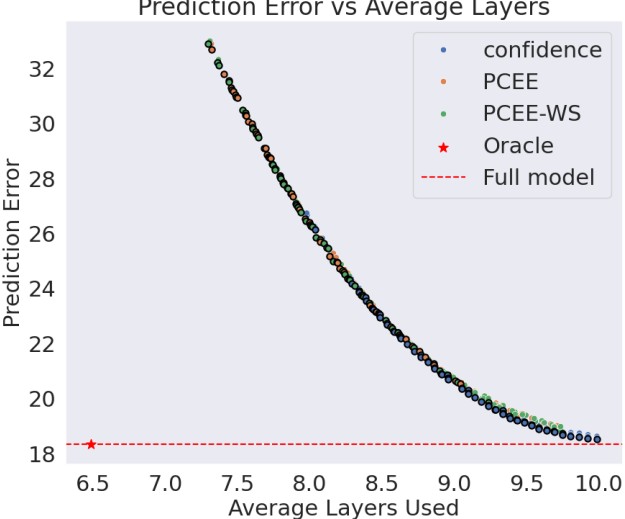

Figure 16: Accuracy/compute trade-off for pre-trained ViT on ImageNet, averaged over 5 runs and with points on the Pareto front circled in black.