# OpenReview forum: "Performance Control in Early Exiting to Deploy Large Models at the Same Cost of Smaller Ones"
_ICLR.cc/2025/Conference — ICLR 2025 Conference Withdrawn Submission_

### Official Review · Reviewer_vdgL · 2024-10-30

**Soundness:** 1
**Presentation:** 2
**Contribution:** 1
**Rating:** 3
**Confidence:** 3

**Summary:**

This paper proposes a simple way to modify the decisions made by early exiting networks. Basically, the predicted “confidence” scores by the exit layer blocks are transformed into estimated accuracy scores, based on a precomputed histogram of mapped values. Since the histogram can be computed from an already-trained early-exit network, the authors describe it as a post-hoc approach. The method is evaluated on CIFAR-100 in the main paper for image classification.

**Strengths:**

I think that improving early exit networks, and more generally, dynamic neural networks, is an important and meaningful direction, and is worthy of investigation.

**Weaknesses:**

The contributions and novelty of the paper seems to be rather far from the level required at ICLR. In particular, early exiting networks have been around for a long time, and simply proposing to better map the predicted “confidence” scores to accuracy based on a pre-computed histogram from a held-out set is rather incremental. In short, the proposed process is very simple -- compute the average accuracy of samples at a certain confidence score, and use the average accuracy as a signal instead of the predicted “confidence” scores. This could be a small trick to use to better calibrate the early exit networks, but definitely not something enough for a full paper.
Since the proposed method is too simple, there is nothing much to comment about it.

Besides this, there are also various serious weaknesses in the experiments. These weaknesses are likely not be able to fixed within the rebuttal window.


Firstly, the number of compared baselines is very small. Only two existing methods are really being compared with. Yet, early exit networks have existed for quite a long time, and there should be a long list of baselines to compare with.

Secondly, the authors also used a single “backbone” early exit network MSDNet (albeit at three different sizes). This baseline is already quite dated, as it comes from a 2017 paper. Besides the fact that only one early exit network “backbone” is used (which is not convincing), the relevance of the experiment results are also in doubt, since they are applied to an old architecture. For instance, do these results transfer well to newer architectures?


Furthermore, only results from one dataset (CIFAR-100) is reported in the main paper. Yet, the reported results in Table 4 on CIFAR-100 does not even show very significant improvements.  For instance, for the accuracy results on MSDNet Small setting, the improvements over TS+Confidence is often not so significant.

Moreover, in Table 4, these results have been reported over a very small range of threshold values (delta), which is 0.65-0.71 for the small model and 0.67-0.73 for the large model. In fact, not only are these threshold values over a small range, they are also different between the different model settings, suggesting that these values have been cherry-picked. These results are not convincing to me. Why have these threshold values been selected? Also, can the authors provide results on a wider range of threshold values?


At the same time, here are some other suggestions to improve the relevance and significance of the paper:

Experiments can be done on more recent early exit networks, such as Transformer networks. For instance, at various parts of the paper, the authors discuss speculative decoding as another way of improving inference efficiency, and speculative decoding is most often applied on Transformers, and also often on text decoding tasks. However, Transformers were not even experimented on at all.

Since the proposed method is very general, it does not need to be only tested on image tasks. For example, it could (and also should) be easily tested on text or video-based modalities as well.

**Questions:**

Please refer to “Weakness” section for the concerns/questions.

---

> ### Author Response · Authors · 2024-11-23
>
> Dear Reviewer,
>
> Thank you for your feedback, we appreciate the time you dedicated to reviewing our work.  Below, we respond to the concerns you raised in your review.
>
> > The proposed process is very simple.
>
> Simplicity in a method should not detract from its value, particularly when the method achieves strong and consistent performance. Overly complex methods can introduce unnecessary overhead, reduce interpretability, and hinder adoption in real-world applications. For instance, Temperature Scaling (TS) exemplifies an exceedingly simple technique—it adjusts logits by dividing them with a single scalar. Despite its simplicity, TS is widely used as a standard calibration method due to its effectiveness and ease of integration into existing workflows [4, 5, 6]. Similarly, our method provides an intuitive and computationally efficient solution for performance control in early exiting, achieving competitive performance while maintaining usability. We believe this aligns with the scientific community’s emphasis on methods that are not only novel but also practical and impactful.
>
> > MSDNet is already quite dated.
>
> We used the MSDNet architecture to ensure fair comparisons with prior work and established baselines such as Laplace. MSDNet's architectural design effectively addresses key challenges in integrating intermediate exit layers into CNNs, such as the lack of coarse features in early layers and gradient conflicts from multiple loss sources. These strengths make it a widely used benchmark, even in recent works published in 2024 [1, 2, 3]. While MSDNet was introduced in 2017, it remains a foundational architecture for early exiting in computer vision.
>
> > The improvements over TS+Confidence is often not so significant.
>
> We discuss this in the *Effect of Calibration* subpart of Section 5 of the paper. The integration of Temperature Scaling (TS) with confidence thresholding enhances performance relative to the confidence baseline, and this is an expected result, as TS improves model calibration, and hence accuracy of predictions when early exiting, and although the difference is not significant, TS results are slightly worse than those of our proposed PCEE and PCEE-WS methods. The important factor to compare is the additional training and hyperparameter tuning required for TS. As highlighted in the related work, model calibration could be a challenging task influenced by various architectural and hyperparameter factors, such as depth, width, and choice of optimizer. Our approach, on the other hand, offers a simpler alternative that does not necessitate any extra training and remains effective in mitigating overconfidence in models.

---

> ### Author Response · Authors · 2024-11-23
>
> > Moreover, in Table 4, these results have been reported over a very small range of threshold values (delta), which is 0.65-0.71 for the small model and 0.67-0.73 for the large model. In fact, not only are these threshold values over a small range, they are also different between the different model settings, suggesting that these values have been cherry-picked.
>
> > Why have these threshold values been selected? Also, can the authors provide results on a wider range of threshold values?
>
> **Figure 6** in the paper illustrates accuracy vs. threshold for a larger set of threshold values than those shown in Table 4. The three thresholds in the table were selected to enhance clarity and provide a concise comparison. Specifically:
>
> - The maximum threshold corresponds to the peak accuracy achievable by the full model, as we cannot provide performance control for thresholds larger than the accuracy of the full model in principle.
>
> - The other two thresholds were chosen with a step size of -0.03 for consistency.
>
> This is why the largest threshold differs between the small and large models. Importantly, as seen in Figure 6, both PCEE and PCEE-WS consistently outperform confidence thresholding while maintaining accuracy above the set threshold, **demonstrating that the thresholds in Table 4 were not cherry-picked**. For better visualization, Figure 6 uses a restricted range of thresholds compared to [0, max_acc]. Nevertheless, the trends clearly validate the robustness of our approach and support the conclusions drawn from the experiments.
>
> ---
>
> [1] Meronen, Lassi, et al. "Fixing overconfidence in dynamic neural networks.", 2024.
>
> [2] Jazbec, Metod, et al. "Towards anytime classification in early-exit architectures by enforcing conditional monotonicity.", 2024.
>
> [3] F. Ilhan et al. "Adaptive deep neural network inference optimization with eenet.", 2024.
>
> [4] Balanya, Sergio A., Juan Maroñas, and Daniel Ramos. "Adaptive temperature scaling for robust calibration of deep neural networks." Neural Computing and Applications 36.14 (2024): 8073-8095.
>
> [5] Frenkel, Lior, and Jacob Goldberger. "Network calibration by temperature scaling based on the predicted confidence." 2022 30th European Signal Processing Conference (EUSIPCO). IEEE, 2022.
>
> [6] Shih, Andy, Dorsa Sadigh, and Stefano Ermon. "Long horizon temperature scaling." International Conference on Machine Learning. PMLR, 2023.

---

> > ### Comment · Reviewer_vdgL · 2024-11-24
> >
> > I thank the authors for their response. However, most of my concerns still remain.
> >
> > For instance, MSD is a baseline that was developed in 2017. Even if the authors experiment on them to compare with some previous works, they should still experiment with more modern architectures, i.e., the following comment has not been addressed: “Besides the fact that only one early exit network “backbone” is used (which is not convincing), the relevance of the experiment results are also in doubt, since they are applied to an old architecture. For instance, do these results transfer well to newer architectures?”
> >
> > Furthermore, for such a simple method, it needs to be very widely tested. Yet, only experiments on one dataset is reported, and the improvements are not even so significant. These issues have also been highlighted by other reviewers.
> >
> > Thus, since most of my concerns remain, I choose to keep my score.

---

### Official Review · Reviewer_5WA2 · 2024-10-31

**Soundness:** 3
**Presentation:** 3
**Contribution:** 3
**Rating:** 5
**Confidence:** 4

**Summary:**

This article proposes an accuracy-based early stopping method called PCEE to address the issue of accuracy degradation caused by overconfidence in confidence-based early stopping methods. PCEE achieves this by constructing a reliability diagram (confidence-to-accuracy mapping) to convert the confidence output at the early stopping point into accuracy, and determines whether to stop early based on whether the accuracy exceeds a threshold.

**Strengths:**

1. This article provides extensive experimental evidence of the existence of overconfidence phenomena in ViT and CNN models.

2. This article proposes a simple yet interesting idea of mapping confidence to accuracy to avoid the issue of overconfidence.

**Weaknesses:**

1. The paper employs a reliability diagram to implement an accuracy-based early stopping mechanism through constructing a confidence-to-accuracy mapping. If that is the case, why not directly output the accuracy of each layer's classification of the input samples as the basis for determining whether early stopping is needed? An ablation study on this would be beneficial.

2. Figure 3 shows the results of overconfidence in a 12-layer ViT; is there a similar phenomenon in CNNs, such as MSDNet?

3. In Table 4, although PCEE achieves better accuracy than the baseline oracle, the Avg Layers and Avg FLOPs it uses seem to be higher than those of the oracle, which does not adequately validate the accuracy of PCEE. In other words, although the accuracy of the Laplace or Confidence methods is lower, their Avg Layers and Avg FLOPs are also lower.

4. If the network does not employ early stopping, what is the output accuracy? How does it compare with the accuracy after implementing early stopping? Is there any loss in precision, and if so, how much?

**Questions:**

See weaknesses.

---

> ### Author Response · Authors · 2024-11-23
>
> Dear Reviewer,
>
> Thank you for your feedback, we appreciate the time you dedicated to reviewing our work.  Below, we respond to the concerns you raised in your review.
>
> > The paper employs a reliability diagram to implement an accuracy-based early stopping mechanism through constructing a confidence-to-accuracy mapping. If that is the case, why not directly output the accuracy of each layer's classification of the input samples as the basis for determining whether early stopping is needed?
>
> Using the overall accuracy of each layer without considering the confidence-to-accuracy mapping would not provide the adaptability needed to make decisions based on individual examples. Based on your suggestion, all samples would exit at the same layer, resulting in reduced compute but compromised accuracy. With a sample-dependent mechanism like PCEE, we can reduce compute without compromising accuracy, leveraging the fact that some samples are easy to correctly classify at an early layer.
>
>
> > An ablation study on this would be beneficial.
>
> **Table 2** shows the accuracy for each layer if we stop at that layer for all examples. You can see from Table 4 that the accuracy of PCEE (73.85%) and PCEE-WS (73.97%) are very close to the accuracy of layer 6 of MSDNet-large (74.17%) although our method uses only 3 layers on average. This shows the superiority of our method upon a fixed-layer exiting due to its adaptive nature.
>
>
> > Figure 3 shows the results of overconfidence in a 12-layer ViT; is there a similar phenomenon in CNNs, such as MSDNet?
>
> Yes, as shown in Figure 4 and in Appendix D of the paper, MSDNet is also overconfident on CIFAR datasets and overconfidence grows with the depth and size of the model. To further clarify, we have included a figure similar to Figure 3 for MSDNet on CIFAR-100 in the updated version of the paper for your reference (see **Figure 10 in Appendix D**). The added parts are highlighted in yellow for better readability. The figure shows that overconfident early layers trigger a (premature) exit on layer 1, the first layer surpassing the given threshold although the model makes incorrect predictions until layer 6.

---

> ### Author Response · Authors · 2024-11-23
>
> > In Table 4, although PCEE achieves better accuracy than the baseline oracle, the Avg Layers and Avg FLOPs it uses seem to be higher than those of the oracle, which does not adequately validate the accuracy of PCEE.
>
> Please note that in Table 4 we included Oracle as a baseline only to show the best achievable computation gain without losing any performance. *Oracle has access to privileged information unavailable to any practical early exit method*—it uses the final layer's prediction for each test example to determine the earliest intermediate layer with matching predictions. This access renders Oracle impractical for real-world scenarios, as access to the final layer’s prediction requires using all the layers and thus defeats the purpose of early exiting. **Therefore, we normally expect PCEE to achieve a lower accuracy (and higher Avg FLOPs) than Oracle**. The fact that PCEE or PCEE-WS sometimes achieve a higher accuracy than Oracle is a surprising observation, indicating that certain intermediate layers can predict the correct label even when the final layer cannot. This suggests a potential regularizing effect of early exiting which is interesting to study for future work.
>
>
> > Although the accuracy of the Laplace or Confidence methods is lower, their Avg Layers and Avg FLOPs are also lower.
>
> We address this trade-off in Section 5 of our paper by highlighting that despite the increase in the average number of used layers and FLOPs, PCEE and PCEE-WS achieve higher performance, potentially justifying the computational trade-offs in situations where accuracy is of priority. The key advantage of PCEE and PCEE-WS lies in their ability to provide performance control, which is highly important for the reasons laid out throughout the paper. Unlike methods requiring exhaustive multi-layer threshold searches, our method uses a simple, user-defined accuracy lower bound, reducing pre-inference costs and allowing easy adjustment of the threshold based on the desired accuracy and budget constraints. While the Laplace baseline achieves the lowest average layers, it falls below the threshold for most of the $\delta$ values and therefore does not provide reliable performance control.
>
>
>
> > If the network does not employ early stopping, what is the output accuracy? How does it compare with the accuracy after implementing early stopping? Is there any loss in precision, and if so, how much?
>
> If the network does not employ early exiting, the output corresponds to that of the full model which is the same as the accuracy of *Oracle* throughout our paper. Oracle uses the final layer's prediction for each test example to determine the earliest intermediate layer with matching predictions and hence has access to privileged information that is unavailable to practical early exiting methods. Therefore, with practical EE methods such as ours (PCEE and PCEE-WS) we incur some levels of performance degradation at the benefit of significant computational savings. Therefore, **we answer your question “Is there any loss in precision, and if so, how much?” in Section 5**, where you can compare the accuracy of PCEE and PCEE-WS with the accuracy of Oracle. As you can see, in most cases, there is a drop in accuracy when we use PCEE or PCEE-WS, but this drop is very small compared to the huge computation gain we get—often requiring only half the layers to deliver accuracy very close to that of the full model.

---

> > ### Comment · Reviewer_5WA2 · 2024-11-27
> >
> > Thank you for your response, which has addressed most of my concerns. However, I still have the following issue: The authors claim that PCEE reduces computational cost compared to Oracle, while the drop in accuracy remains acceptable. A similar phenomenon is observed with TS+Confidence, where the decrease in accuracy is also tolerable. This seems insufficient to fully demonstrate the superiority of PCEE.

---

> ### Author Response · Authors · 2024-12-04
>
> Dear Reviewer,
>
> Thank you for taking the time to read our rebuttal. We are glad to have addressed most of your concerns.
>
> We would like to clarify that we do not claim that PCEE reduces computational cost compared to Oracle. In fact, PCEE typically incurs a higher computational cost and lower accuracy than Oracle. However, as detailed in our previous response, Oracle is not a practical baseline because it relies on privileged information (the last layer’s predictions) that is unavailable to real-world early exit methods. To see the superiority of PCEE and PCEE-WS, they should be compared to our practical baselines, such as Confidence and Laplace. Our method outperforms these baselines in maintaining performance control, which is the primary goal of our approach. Also, regarding the comparison with TS+Confidence, while its results are comparable to those of PCEE and PCEE-WS, TS introduces additional complexity. It requires extra training and hyperparameter tuning, which depend on architectural and optimization choices, making calibration more challenging. In contrast, PCEE avoids these issues as it requires no additional training and effectively mitigates model overconfidence. For further details, please refer to the “Effect of Calibration” subpart of Section 5 of the paper.
>
> We hope our responses have sufficiently addressed your concerns. If so, we would greatly appreciate your consideration in raising your score.

---

### Official Review · Reviewer_H7hX · 2024-11-02

**Soundness:** 2
**Presentation:** 3
**Contribution:** 2
**Rating:** 3
**Confidence:** 5

**Summary:**

The paper introduces PCEE (Performance Control Early Exiting), a method that ensures a lower bound on accuracy by thresholding based not on a datapoint’s confidence but on the average accuracy of its nearest samples from a held-out validation set with similar confidences. This approach offers a simple yet computationally efficient alternative that provides control over performance, facilitating
accurate adaptation of EE methods for practical use. Comparisons have been carried out with the existing baselines on CIFAR-10 and CIFAR-100.

**Strengths:**

- The paper is clear and well-written
- The initial analysis regarding the model accuracy and confidence is informative.

**Weaknesses:**

- Authors mention that the proposed method is evaluated on the samples with similar confidence from a held-out dataset. How is it ensured that the held-out dataset has similar confidence trends as the test sample ?
- Is PCEE evaluated at each layer? How much computation overhead does it add in terms of time? Did the authors try to employ PCEE on some selected layers only?
- The application scope of the paper is limited. The authors have tested the approach in classification settings, that too only on 2 datasets. It is important to verify the effectiveness on other deep learning tasks as well.
- From the Table 4, the performance of the proposed method does not seem to be consistent as the baselines seem to be overperforming in many cases.
- Overall, while the paper is explained nicely, there are major issues regarding the scope of application on more tasks and datasets and subpar results.

**Questions:**

Please check the weaknesses section.

---

> ### Author Response · Authors · 2024-11-23
>
> Dear Reviewer,
>
> Thank you for your feedback, we appreciate the time you dedicated to reviewing our work.  Below, we respond to the concerns you raised in your review.
>
> > Authors mention that the proposed method is evaluated on the samples with similar confidence from a held-out dataset. How is it ensured that the held-out dataset has similar confidence trends as the test sample?
>
> If the test and validation datasets follow the same distribution, their confidence score distributions are also expected to be similar. This assumption of in-distribution test and validation data is standard in many supervised learning tasks. However, as mentioned in our Discussion section, PCEE offers opportunities to address discrepancies between validation and test distributions. To address this concern, we note the following:
>
> - This issue can be mitigated in deployment by **continuously updating the reliability diagrams** using fresh data as they come to account for distribution shifts over time. This approach provides a simple online fix to handle discrepancies between validation and test datasets.
>
> - It is worth noting that this limitation is not unique to our method but is shared by other calibration techniques, such as Temperature Scaling, which also rely on validation datasets for calibration. Studies have shown that these techniques are generally ineffective for calibration under distribution shifts [1, 2, 3].
>
> > Is PCEE evaluated at each layer? Did the authors try to employ PCEE on some selected layers only?
>
> Thank you for this suggestion. We evaluated all baselines and PCEE by applying them to all layers, as standard in the literature and because it does not add significant computational overhead (see the following response). For some settings and datasets, it might be more optimal to apply early-exiting to a subset of selected layers, such as by always going through the first $n$ layers of the model. However, selecting the optimal subset of layers for a setting is not straight-forward: if done with heuristics, there is a risk of degrading the overall performance; if done with an adaptive mechanism, e.g. with a routing function, it requires additional training and also comes with a computational cost.
>
> > How much computation overhead does it add in terms of time?
>
> The exit layers we used are only single linear transformations that convert the representation to a vector with the size of the number of classes, so the computational overhead is significantly cheaper than a single layer (block) of the model, which contains several convolution layers in MSDNets and attention layers in ViT. The overall computational cost of the exit layers is minimal, especially when considering the significant computation savings from early exiting. For instance, as shown in Table 4 of our paper, using approximately half of the model layers, PCEE achieves an accuracy very close to that of the full model. Please note that all other baselines also use these exit layers.
>
> ---
>
> [1] Yaniv Ovadia et. al., Can you trust your model's uncertainty? Evaluating predictive uncertainty under dataset shift. NeurIPS 2019.
>
> [2] Muthu Chidambaram et. al, On the Limitations of Temperature Scaling for Distributions with Overlaps, In The Twelfth International Conference on Learning Representations (ICLR) 2024, Vienna, Austria, May 7-11, 2024
>
> [3] Keigo Tada and Hiroki Naganuma, How Image Corruption and Perturbation Affect Out-of-Distribution Generalization and Calibration, In 2023 International Joint Conference on Neural Networks  (IJCNN), pp.1–6, 2023.

---

> ### Author Response · Authors · 2024-11-23
>
> > The application scope of the paper is limited. The authors have tested the approach in classification settings, that too only on 2 datasets. It is important to verify the effectiveness on other deep learning tasks as well.
>
>
> We acknowledge that our experiments are currently limited to classification tasks; however, classification remains a foundational and widely used application in production settings across various domains, making it a practical and impactful domain for evaluating methods like ours. The results on the selected datasets demonstrate the potential of our approach in scenarios where performance control and computational efficiency are critical.
>
> In Section 3, we also present an interesting observation regarding the benefits of increasing model size coupled with early exiting. Specifically, scaling up model size can enhance computational efficiency by enabling early exiting during inference, demonstrating the broader applicability of our method within classification tasks.
>
> We agree that expanding the scope of our approach to other deep learning tasks such as autoregressive language modeling, where computation costs can grow significantly with sequence length, is a promising avenue for future work. We are excited about the potential of our method in such tasks and aim to explore these avenues in subsequent research.
>
>
> > From the Table 4, the performance of the proposed method does not seem to be consistent as the baselines seem to be overperforming in many cases.
>
> Please note that in Table 4 we included Oracle as a baseline only to show the best achievable computation gain without losing any performance. **Oracle has access to privileged information unavailable to any practical early exit method**—it uses the final layer's prediction for each test example to determine the earliest intermediate layer with matching predictions. This access renders Oracle impractical for real-world scenarios, as access to the final layer’s prediction requires using all the layers and defeats the purpose of early exiting. Also, note how Oracle is not dependent on $\delta$, and the numbers are the same for all $\delta$ values. With this explanation, we do not expect PCEE or PCEE-WS to outperform Oracle. It sometimes does, which is a surprising observation, indicating that certain intermediate layers can predict the correct label even when the final layer cannot and suggests a potential regularizing effect of early exiting which is interesting to study for future work.
>
>
> In Table 4, we bold the top two accuracies, which often include Oracle---as it has the accuracy of the full model---and one of our methods. Importantly, our method outperforms practical baselines such as Confidence, Confidence + Temperature Scaling (TS), and Laplace while maintaining performance control, which is the primary goal of our approach. Furthermore, our method can be effectively combined with Temperature Scaling to enhance performance further.

---

> > ### Comment · Reviewer_H7hX · 2024-11-24
> >
> > I appreciate the authors for a detailed response. However, there are still concerns regarding the proposed work:
> >
> > - The method is utterly simple and to effectively validate its effectiveness, a thorough analysis is required across many architectures , datasets, tasks and modalities. Currently the paper is in its early stage, slightly below the ICLR standards and hence I encourage authors to perform thorough analysis and submit to some other venue.
> >
> > - I acknowledge the oracle settings, however, my comment was directed towards the performance of proposed approach with other baselines. The proposed approach performs slightly better than the best baselines and scope of comparisons are limited.
> >
> > - Hence, at this stage, the paper requires more work.

---

### Note · Authors · 2025-01-22

I have read and agree with the venue's withdrawal policy on behalf of myself and my co-authors.